# Signatures of Dermal Fibroblasts from RDEB Pediatric Patients

**DOI:** 10.3390/ijms22041792

**Published:** 2021-02-11

**Authors:** Arkadii K. Beilin, Nadezhda A. Evtushenko, Daniil K. Lukyanov, Nikolay N. Murashkin, Eduard T. Ambarchian, Alexander A. Pushkov, Kirill V. Savostyanov, Andrey P. Fisenko, Olga S. Rogovaya, Andrey V. Vasiliev, Ekaterina A. Vorotelyak, Nadya G. Gurskaya

**Affiliations:** 1Center for Precision Genome Editing and Genetic Technologies for Biomedicine, Pirogov Russian National Research Medical University, Ostrovityanova 1, 117997 Moscow, Russia; arkadii.beilin@gmail.com (A.K.B.); hopeevt@gmail.com (N.A.E.); 2Koltzov Institute of Developmental Biology of Russian Academy of Sciences, 26 Vavilova Str., 119334 Moscow, Russia; rogovaya26f@gmail.com (O.S.R.); 113162@bk.ru (A.V.V.); vorotelyak@yandex.ru (E.A.V.); 3Center of Life Sciences, Skolkovo Institute of Science and Technology, Bolshoy Boulevard 30, Building 1, 121205 Moscow, Russia; lukyanovd7@gmail.com; 4Shemyakin-Ovchinnikov Institute of Bioorganic Chemistry, Miklukho-Maklaya 16/10, 117997 Moscow, Russia; 5National Medical Research Center for Children’s Health, Federal State Autonomous Institution of the Ministry of Health of the Russian Federation, Lomonosovsky Prospekt, 2, Building 1, 119296 Moscow, Russia; m_nn2001@mail.ru (N.N.M.); edo_amb@mail.ru (E.T.A.); pushkovgenetika@gmail.com (A.A.P.); 7443333@gmail.com (K.V.S.); director@nczd.ru (A.P.F.); 6Diagnostic and Treatment Department, University Children’s Hospital (UCH), Sechenov First Moscow State Medical University (Sechenov University), 119435 Moscow, Russia

**Keywords:** epidermolysis bullosa, RDEB, COL7A1, disease mutation, splicing, dermal fibroblast, differential gene expression, cell culture, extracellular matrix

## Abstract

The recessive form of dystrophic epidermolysis bullosa (RDEB) is a debilitating disease caused by impairments in the junctions of the dermis and the basement membrane of the epidermis. Mutations in the COL7A1 gene induce multiple abnormalities, including chronic inflammation and profibrotic changes in the skin. However, the correlations between the specific mutations in COL7A1 and their phenotypic output remain largely unexplored. The mutations in the COL7A1 gene, described here, were found in the DEB register. Among them, two homozygous mutations and two cases of compound heterozygous mutations were identified. We created the panel of primary patient-specific RDEB fibroblast lines (FEB) and compared it with control fibroblasts from healthy donors (FHC). The set of morphological features and the contraction capacity of the cells distinguished FEB from FHC. We also report the relationships between the mutations and several phenotypic traits of the FEB. Based on the analysis of the available RNA-seq data of RDEB fibroblasts, we performed an RT-qPCR gene expression analysis of our cell lines, confirming the differential status of multiple genes while uncovering the new ones. We anticipate that our panels of cell lines will be useful not only for studying RDEB signatures but also for investigating the overall mechanisms involved in disease progression.

## 1. Introduction

Up to twenty-four genetic subtypes of epidermolysis bullosa (EB) have currently been described for the four main types of the disease. The mutations can be used to identify an inheritance type and predict the further course of the disease. Dystrophic epidermolysis bullosa (DEB) is one of the most severe forms of EB. The skin tissue of DEB patients displays anomalies of anchoring fibrils (AF), blistering and separation of sublamina densa as a result of mutations in the *COL7A1* gene [1]. The mutation in *COL7A1* could be of two forms: a biallelic pathogenic variant (recessive form of DEB, RDEB) or a heterozygous pathogenic variant (dominant form of DEB). Here, we focus on RDEB, which includes forms that vary in the severity of disease manifestations. Severe cases exhibit lifelong erosions and blistering that occur in the skin and often in the gastro-intestinal tract, leaving scars after healing. Other clinical manifestations include mitten-like deformities and the loss of nails. There are also less severe subtypes of RDEB with localized skin lesions or mucosal involvement. The pain and itching are most often experienced by RDEB patients, and skin squamous carcinoma often arises in patients aged 30 to 50 years [2]. To date, more than 800 different mutations have been identified in the *COL7A1* gene, localizing in different parts of the sequence and leading to cutaneous manifestations of varying severity. However, it is unknown whether the whole spectrum of RDEB symptomatic variants could come down to the type of mutation in the *COL7A1* gene [3].

Type VII collagen (C7) is secreted by both epidermal keratinocytes and dermal fibroblasts. Dermal fibroblasts represent a major component and a target of permanent inflammation. They are also involved in the development of granulation tissue and scars connected with regenerative processes. The dysregulated gene expression in RDEB fibroblasts leads to the acquisition of traits which drive the development of skin squamous carcinoma [4]. Hence, dermal fibroblasts of DEB patients represent an appropriate model to investigate the pro- and antifibrotic processes and their connection with prolonged inflammation during the rounds of skin regeneration.

The new strategies for targeted RDEB therapy are being developed, but still face significant challenges in their clinical applications [5,6]. Revealing mechanisms that provide an effective antifibrotic interference or prevent cancer-associated transformation could be helpful in advancing the approaches for RDEB therapies [7,8,9,10]. A model characterizing the stages of fibrogenesis could also be effective for the evaluation of the new therapeutic inhibitors of the fibrotic cascade.

The goal of this study was to obtain, characterize and compare the properties of the primary dermal fibroblast cultures from RDEB and healthy patients. All RDEB patients had OMIM entry 120120 but showed different clinical manifestations. The mutations in the *COL7A1* gene from four patients were found by Next Generation Sequencing NGS and confirmed by Sanger sequencing. The expression of *COL7A1* was assessed in both groups of fibroblast cultures. We also investigated the possible correlation between the expression pattern of cells with the mutation in *COL7A1* and several of their phenotypic properties in culture, including cell morphology and the contraction of extracellular matrix (ECM).

We report a set of differentially expressed genes (DEG) between the RDEB and healthy fibroblasts and compare them with DEG obtained from the publicly available RNA-seq analysis of RDEB fibroblasts. The profile of DEG as the result of disease progression depends on the genetic context, which includes the nature of COL7A1 mutation, Single Nucleotide Polymorphism SNPs of other genes and genetic background. The important aspect of this study is to find the specific RDEB patterns of DEG not only in our FEB lines but also from the study of other RDEB fibroblasts lines, i.e., with the *c.6527insC* COL7A1 mutation. The comparison of these transcriptomes with the data of our FEB lines will help to decipher the common traits of differential expression in RDEB across different genetic backgrounds. 

## 2. Results

### 2.1. Description of the Experimental Material

The four RDEB subjects were male, ranging from 3 to 21 years old and originating from different parts of the Russian Federation (the Kaliningrad region, the Ulyanovsky region, Dagestan, the Moscow region). The seven healthy patients included three males, ranging from 8 to 58 years old and originating from the Central region of the Russian Federation. Cell lines obtained from RDEB and healthy subjects are referred to as FEB1–FEB4 (fibroblasts of epidermolysis bullosa patient, FEB) and FHC1–FHC7 (fibroblasts of healthy control donor, FHC) correspondingly (Table 1).

### 2.2. State of Health and Genetics of Patients d1–d4

RDEB patients that participated in the study (d1–d4) displayed overlapping but not identical sets of phenotypic features. Patient d1 had the severe generalized subtype of RDEB, characterized by generalized skin pathological traits, represented by spots, blisters, milia, erosion, scales, crusts and scars, dystrophy (or absence) of hands and feet nail plates as well as slight pseudosyndactyly. A similar pattern of pathology was determined for patient d3, with minor differences in the degree of pathological trait expression (Appendix A). Patient d2 displayed blistering on the trunk and the neck without the severe, mutilating scarring, which is indicative of the rare form of RDEB inversa [11]. The health status of this subject was similar to those described in a previous study [12]. The skin lesions represented erosions that form atrophic scars and milia after healing. The intertriginous skin sites were in the abdomen area, and the sites at the base of the neck and the lumbar spine were covered with white scars at the places of the previous erosions. There were multiple erosions present on the mucosa, as well as the progressive restriction of oral aperture and microstomy. Partial and fully developed pseudosyndactyly were on the hands and on the legs, respectively, with no nail plates on the legs. Patient d4 with a generalized intermediate subtype of RDEB also showed blisters and erosions with affected oral mucosa. Pseudosyndactyly, foot deformity, joint contractures and microstomia were observed. In summary, patients d2 and d4 were affected less severely than patients d1 and d3, who showed a harsher clinical manifestation of RDEB, although no patients showed the signs of skin squamous carcinoma development.

### 2.3. Mutations in the COL7A1 Gene

The RDEB mutations described in this paper can be divided into two groups: splice site mutations and missense mutations. In the ClinVar and Human Gene Mutation Database (HGMD), the mutation of studied patients fell into the EB category of dystrophic type with the autosomal recessive inheritance (Table 1).

Target regions of the COL7A1 gene with the mutations leading to DEB symptom complexes were investigated by massive parallel sequencing and confirmed by Sanger sequencing. Patient d1 had a missense variant c.425A > G (p.K142R, rs121912856) in a homozygous state (Appendix A). This transition is common among patients of the European population with DEB (HGMD Professional 2012.1) [13]. The heterozygous state of this mutation was also described, manifesting in the mild form of the disease, RDEB localisata [14]. Patient d2 represents the case of compound heterozygous mutation composed of the intronic mutation c.682 + 1G > A and missense variant c.6205C > T (p.R2069C, rs121912855) in exon 74 of the COL7A1 gene (Appendix A). The c.6205C > T variant has been described in a few cases of the RDEB inversa subtype in the Iranian and Japanese populations [12,13,14]. These mutations were quite frequently detected in the European population, both separately and in the combination with c.425A > G [15,16,17]. Patient d3 represents the rare case of the combination of the RDEB pathological mutation in exon 111 c.8245G > A (p.G2749R, rs121912853) in a homozygous state (ClinVAR accession VCV000017460) with the second heterozygous mutation in the KRT5 of c.1054C > T p.R352C (Appendix A). The similar mutation of keratin 5, R352S, was described as the first mutation in the 2A domain of KRT5 for Japanese and Korean EBS patients [18]. It is also worth noting that patient d3 was born from a consanguineous marriage, although the pedigree of the other RDEB generalized severe patient d1 is not known.

Patient d4 had the c.520G > A mutation in exon 4 of COL7A1. This pathological variant was viewed as a glycine missense mutation, p.G174R [19]. We assumed the splicing impairment in this case. For prediction of splicing, we used the Splice Site Prediction by Neural Network (NNSPLICE) (Appendix A).

#### RT-PCR Analysis for Verification of COL7A1 Splice Defects

The disturbances of splicing predicted in silico (Appendix A) were verified by amplification with the sets of specific primers on the templates of respective cDNA samples. We have shown the appearance of additional fragments in FEB1 and FEB4 cells (Figure 1A,B). For FEB1, two different fragments corresponding to COL7A1 transcripts were revealed with each pair of specific primers (Figure 1A, lanes 2 and 6). The sequencing of the bands confirmed the existence of transcripts with intron 3 retained. Moreover, the normal spliced variant that contained the c.425A > G mutation was also shown to exist (Appendix A). The splicing impairment in the FEB2 line (Appendix A) was not detected by RT-PCR (Figure 1A), and the sequencing of cDNA revealed only normally spliced transcripts. RT-PCR on the FEB4 template showed the additional band predicted to exist as a result of the mutation in the splice site (Figure 1B).

### 2.4. Immunohistochemistry of Skin Biopsy (IHC)

Skin biopsies from patients d6, d3 and d2 were studied for C7 expression (Figure 2A–C). Whereas in the healthy skin, C7 is located around the dermoepidermal junction, in the skin of the RDEB patients, the expression of C7 fails to relocate to the sublamina densa, instead spreading across the dermis in the skin. The negative control of immunostaining demonstrated that the Immunohystochemistry IHC test is specific (Figure 2D). The intensity of the signal was almost not reduced in the case of d2 (Figure 2B) and mildly reduced in d3 (Figure 2C). The signal integrity breach and delocalization indicated that the AF function is disrupted. This finding is consistent with previously reported data [20] on a recombinant mutant with the similar mutation p.G2749R C7, who was capable of the secretion but not production of the mature trimeric form of C7.

The mutations in FEB1, FEB2 and FEB4 are localized within the cartilage matrix protein homology region of C7, which is located upstream of the epitope for the LH7.2 antibody. The epitope was considered to be within the FN3 domain of C7, which is encoded by the downstream exons relative to exons 3–6 affected in FEB lines [21].

Noncollagen NC1 domain with matrix binding sites was considered the most immunoreactive part of C7 [21]. The use of monoclonal LH7.2 antibody for skin IHC allows confirmation of the different expression pattern and AF dysfunction in the case of patient d3 (Figure 2).

### 2.5. Characterization of Fibroblast Cell Lines

All primary dermal fibroblasts were checked for the presence of mycoplasma, and Short Tandem Repeat STR profiling was made. No mycoplasma or cross-contamination were found. A cytogenetic study confirmed the absence of chromosomal aberrations after 10 passages.

#### 2.5.1. Fibroblast Markers’ Expression

Fibroblasts from both FEB and FHC lines were analyzed for the expression of the fibroblasts markers S100A4, transgelin (TAGL), collagen IV, collagen I, fibronectin, alpha-smooth actin (αSMA), ED-A segment of fibronectin (FN ED-A) and type VII collagen (Figure 3). Cells showed the equal expression of markers with the exceptions of C7, α-SMA and FN ED-A. We applied the method of semi-quantitative comparison of Immunocytochemistry ICC data of the C7, αSMA and FN ED-A expression levels. 

Staining intensity for C7 was measured for FEB (1–3) and FHC (1–4). The means of these groups were compared by nested *t* test. The results show that FEB cells have lower C7 staining intensity compared to FHC cells (*p* < 0.001) (Appendix A; Appendix A).

Staining intensities for α-SMA and FN ED-A were measured for FEB (1–4) and FHC1. Samples were compared using one-way ANOVA with post-hoc Tukey HSD test. Expression levels of α-SMA and FN ED-A of every FEB line were significantly (*p* < 0.05) higher than those of FHC (Figure 3C,D).

#### 2.5.2. Morphology Analysis

All primary dermal fibroblasts showed a normal spindle-like morphology (Figure 4A). For morphological analysis, the number of measured cells ranged from 43 to 127 for each cell line. To compare the FEB and FHC, we used combined morphology data from FEB (1–4) and FHC1. Cell shapes were compared by the two-sample *t* test. Flow cytometry Forward Scatter (FSC) and Side Scatter SSC data were compared by the FlowJo chi squared comparison. FHC1 was compared to FEB (1–4) duplicates. Baseline T(x) was obtained by comparing FHC1 to the FHC1 repeat plus FHC3 in duplicate.

The results revealed that FEB cells on average have a bigger area, perimeter, minimum fit ellipse axis and minimum caliper diameter. Also, FEB cells were shown to be less circular and solid (Figure 4B; Table 2). No significant difference in SSC between combined FEB and FHC was found, but FSC comparison showed that FEB cells on average are slightly larger (T(x) = 868, baseline T(x) = 215) (Figure 4C; Table 2).

#### 2.5.3. Contraction of Collagen Gel

The contraction of four collagen gels was measured for each FEB (1, 2, 3 and 4) and FHC1. The final data was combined into two groups (FEB and FHC) and compared by Mann–Whitney test. The results show that fibroblast-induced collagen contraction was significantly higher in FEB cells in comparison with FHC cell lines (*p* < 0.05). The smaller the area of the gel, the higher the cell contraction recorded (Figure 4D; Table 2).

#### 2.5.4. Western Blot of Collagen VII

The C7 content in fibroblast lines was evaluated by immunoblot of cellular lysates with antibodies specific for the non-helical terminal region of type VII collagen. Lysates of the FHC line were used as a control for normal content of protein. β-actin as a loading control was used for semi-quantitative estimation of C7. Lower C7 content was demonstrated in FEB (1–4) lines in comparison with those of FHC (1 and 2) lines (Figure 4E). The presence of C7 in each of the FEB samples may account for the more or less intact part of the N-terminal domain of C7. Here, we used the antibodies against the synthetic peptide TVQYSDDPRTEF, which corresponds to the fragment of amino acid residues 81 to 92 of the NC1 domain of C7. This fragment is intact in all FEB mutants of C7. The diminished level of C7 protein has been demonstrated for all FEB lines irrespective of the mutation site (Figure 4E). Due to the heterozygous state of the mutation, FEB2 and FEB4 were able to produce a minor portion of the protein with the N-terminal domain of C7 being more or less intact. In FEB1, the low amounts of normally spliced mRNA could be the cause of such residual immunostaining. While the homozygous mutation present in the FEB3 line kept the structure of the NC1 domain intact, Western Blot WB showed a lack of C7 in FEB3 (Figure 4E).

### 2.6. Transcriptome Analysis

The evaluation of DEGs in RDEB fibroblasts cell lines against healthy controls could be used for the investigation of the key factors in disease progression.

To find the common and specific traits of DEG patterns in RDEB, we focused on the RNA-seq data of other fibroblasts induced by the c.6527insC mutation that was found to be recurrent in the Spanish RDEB population [22]. This mutation induces a premature stop codon (PTC) in exon 80 of COL7A1 and degradation of mRNA that proved to be the reason for the lack of anchoring fibers in the skin of patients [23].

We analyzed RNA-seq data from the study of RDEB fibroblasts from patients who were of different age and sex but had the same mutation, c.6527insC [24]. Publicly available RNA-seq data [25] was used. Differential expression analysis revealed 1567 genes with FDR < 0.05, of which 792 were down-regulated and 775 were up-regulated. Out of the differential expression list, we selected the genes with logFC > 1.5 and logFC < 1 for the gene ontology (GO) analysis (Appendix A). Consistent with the existing literature on RDEB, the results of the PANTHER GO biological processes (BP) overrepresentation test showed terms associated with the regulation of the semaphorin–plexin signaling pathway, ECM organization, cell matrix adhesion and cell adhesion mediated by integrin, smooth muscle cell differentiation, negative regulation of the Notch signaling pathway, regulation of transforming growth factor beta production and epithelial cell proliferation (Appendix A).

### 2.7. RT-qPCR Analysis for DEG of FEB and FHC

Our assessment of the differential gene expression began with the evaluation of several genes known to be involved in the expression profiles of RDEB (Figure 5A,B, Appendix A). We delineated the following list of genes for RT-qPCR analysis, which included, the participants in the C7 process, the components of the dermo-epidermal junction (DEJ), the ECM constituents and the cell adhesion. We also chose to analyze the genes that encode factors which directly participate in inflammation or are regulated in the inflammatory cascade and genes that encode factors associated with pro- and antifibrotic changes.

We determined the diminished expression of transglutaminase 2 (TGM2) in all FEB lines and the reduced level of COL4A1 in all FEB lines excluding FEB3 (Appendix A). We found the upregulation of angiotensin II receptor type 1 (AGTR1), dermatopontin (DPT), cathepsin Z (CTSZ) and AGTR1 in FEB lines compared to FHC (Figure 5A). We obtained new data about the expression of cytokine interleukin 1 beta (IL1B), epiregulin (EREG) and desmoplakin (DSP). Tissue inhibitor of metalloproteinases 3 (TIMP3) had a downregulated expression, as did SMAD family member 7 (SMAD7) (Appendix A). SMAD7 is a known negative regulator of the cascade driven by transforming growth factor β (TGF-β). To evaluate the differential expression of fibronectin (FN1), the alternatively spliced form with the ED-A+ exon (FN-209) was selected for RT-qPCR analysis. Slight downregulation of this form was demonstrated in FEB samples, most clearly seen in FEB1 (Appendix A). S100A4 expression did not differ much from the control lines in FEB1 and FEB3 but was decreased significantly in FEB2 and FEB4 (Figure 5A). Interestingly, only the FEB4 line had increased expression of both alpha-smooth muscle actin (αSMA) and transgelin (TAGL) myofibroblasts markers, while FEB1 showed downregulation of these genes (Appendix A). FEB2 was distinguished from other FEB lines by the upregulation of genes encoding tenascin C (TN-C), fibroblast activation protein alpha (FAP1) and amine oxidase copper-containing 3 (AOC3) and the downregulation of TAGL (Appendix A, Figure 5B). 

As noted before, FEB1/3 are associated with a more severe form of RDEB, while FEB2/4 display the milder phenotype. Curiously, the expression pattern for several genes was different between the pairs of FEB1/3 and FEB2/4, which could reflect the severity of the disease (Figure 5A, dark grey and light grey colors). These include peroxisome proliferator activated receptor gamma (PPARG), decorin (DCN), interleukin 7 (IL7), C-X-C motif chemokine receptor 4 (CXCR4), gliomedin (GLDN), cathepsin B (CTSB), metalloreductase six-transmembrane epithelial antigen of prostate 4 (STEAP4) and melanotransferrin (MELTF) (Figure 5B, Appendix A). We identified several functional clusters of these genes (FEB1/3 vs. FEB2/4): (1) decreased antifibrotic genes (PPARG, DCN, IL7), (2) decreased profibrotic genes (CXCR4, GLDN, CTSB), (3) upregulated TIMP3 and (4) decreased expression of two genes involved in iron uptake and homeostasis (STEAP4 and MELTF) (Figure 5B). Evidently, members of both pro- and antifibrotic cascades are downregulated in FEB1/3 compared with FEB2/4.

PANTHER GO BP analysis of all DEG found by RT-qPCR showed the enrichment of several terms associated with the positive regulation of cell adhesion and apoptosis, regulation of fibroblast proliferation and smooth muscle cell proliferation, negative regulation of locomotion, ECM organization and ion homeostasis (Appendix A). The majority of the DEGs were associated with the extracellular region and cell periphery.

## 3. Discussion

The data obtained in this study could be discussed in several aspects: the role of uncovered mutations in the COL7A1 gene structure and functioning; and the properties of FEB cells, including their functional morphological analysis and patterns of differential gene expression.

C7 damage induced by mutations represents the most essential part of RDEB pathogenesis. Initially, C7 was considered as the important structural protein, a major component of the DEJ, although now it is clear that the general pattern of this pathology’s development is more complicated [26]. The loss of normal ligands of C7 in RDEB causes the skin cells to secrete inflammatory factors as well as matrix and lysosomal proteases [27,28]. In general, the expected traits of RDEB fibroblast expression will be the three interrelated patterns—inflammation, fibrosis and onco-transformation [26,27,28].

Among the four patients with RDEB described here, patients d1 and d3 demonstrated the more severe forms of disease, while d2 and d4 had milder phenotypes. It is important to note that d1/d3 had homozygous states of mutations, while d2/d4 had heterozygous mutations (Table 1).

The COL7A1 mutations of patients d1 and d3 were located in different parts of the gene, disturbing the NC1 domain and distal part of the Triple-Helical Domain THC domain, respectively. However, the generalized severe form of RDEB is characteristic for both patients (Appendix A).

We have shown the compound heterozygous status of mutations in COL7A1 of patient d2 (Table 1). The accumulation of data on d2’s mutations were compared with those previously published and allow us to suggest that the main impact in the RDEB inversa phenotype in d2 is due to the mutation in the exon 74 p.R2069C [29].

Cases of patients with compound heterozygous mutations have been described, including p.G174R, which was found in patient d4. Although it was localized in the NC1 domain of COL7A1, the patients were shown to have a minor expression of functional COL7A1 with rudimentary-appearing AF [15].

FEB1, FEB2 and FEB4 lines possessed pathological COL7A1 mutations with splicing disruption in different exons in the first part of the gene. Splicing impairment dramatically influences the encoded NC1 domain structure and attenuates the expression level of the protein (Figure 1 and Figure 4, Appendix A). Moreover, the destruction of the donor splice site has been assumed to be the main cause of impairments in the cases of FEB1, FEB2 and FEB4 (Appendix A). The most frequent phenomenon of splice site mutation is the appearance of a PTC in an open reading frame, which in the majority of cases leads to the degradation of mRNA due to Nonsense Mediated Decay NMD [5,30]. The degradation of PTC-containing aberrant transcripts may serve as a cellular defense mechanism against the translation of toxic proteins and may be the cause of the pathological deficit of the necessary transcript [31].

The mutations c.425A < G, c.682 + 1G > A and their compound combinations are rather frequent, particularly among the European population of RDEB [16,32], and thus it is important to understand the consequences of a splicing breach. It was demonstrated earlier that the cryptic site activation in _COL7A1_ c.425A < G results in the appearance of a shorter transcript with the skipping of exon 3 [33]. We found that in addition to NMD-sensitive templates, NMD-stable mutant variants of COL7A1 exist in the FEB1 line (Figure 5A, Appendix A). Besides, the unusual phenomenon of normal splicing has been shown to take place in spite of the donor splice site mutation c.425A < G (Figure 5A, Appendix A). Thus, in this case, a “leaking” abnormal splice site may be associated with the synthesis of correctly spliced mRNA, though in a small percentage. This finding underlines the significance of the local context in the areas of impaired splicing sites and may account for the existence of trace amounts of C7 in FEB1 cells in spite of mutation homozygosity. FEB2 carries the heterozygous mutation c.682 + 1G > A in the first nucleotide of intron 5. Despite the impairment in the donor splice site in exon 5, FEB2 displayed only the product of normal splicing (Figure 5A). We briefly consider the influence of the local context of mutation and activity of NMD in the Appendix A.

FEB3 possesses the homozygous missense variant in the downstream exon encoding the part of the THC domain. Moreover, among the patients, d3 had a tendency to stand out because the mutation (G2749R) did not lead to a disturbance of splicing or an interruption of the open reading frame. The mutation was located in the C-end of THC domain. It is important to note that the NC1 domain ligands, most of the THC domain and the NC2 domain cleavage sites were intact. This glycine substitution in C7 did not change the capacities of cell migration and binding with the ECM [34] (Appendix A).

In our study, no significant differences were found in the amount of COL7A1 mRNA content between RDEB and healthy patients, with FEB1 being the only exception (Figure 5A). The levels of COL7A1 mRNA or protein in the cells of patients did not correlate directly with the severity of the disease [35,36,37]. The upregulation of the SP1 and SP5 profibrotic transcription factors could compensate for the low level of COL7A1 mutant transcripts by regulating its transcription.

Based on ICC and WB results, we conclude that the trace amounts of C7 were present in all FEB lines. In FEB1 and FEB3, the level of protein was extremely low despite the different types of mutations. We hypothesize that these patients could be immunotolerant to a therapeutic infusion of recombinant C7, plausible skin engraftment or the most promising bone-marrow transplantation treatments [38,39].

Collagen I, collagen IV and fibronectin are the components of anchoring fibrils and can bind to NC1 domains of C7 [40]. We suggested that C7 defects in RDEB fibroblasts could cause changes in the expression patterns of these proteins. We showed significant reduction in C7 staining intensity for all FEB lines (Figure 3A, Appendix A, Appendix A). No difference was observed for the expression of collagen I and IV. Semi-quantitative analysis showed the increased expression of an alternative form of fibronectin, FN ED-A (Figure 3D), but no difference in total fibronectin expression. This difference was observed in low-density seeded fibroblasts (Figure 3B), but the situation in a dense population could be different [41].

Large scale in vitro experiments with fibroblasts from EB patients of different subtypes demonstrated the extremely broad range of contractility (normal, poor and hypercontraction) [42]. We have shown that on average, FEB lines have a higher contraction rate than healthy cells (Figure 4D). The enhanced contraction of FEB may indicate the profibrotic processes in the dermis of RDEB patients, which should be taken into consideration when we discuss the expression differences of genes associated with profibrotic and antifibrotic properties (Table 2). We tested whether the enhanced contraction of FEB cells may be explained by the heightened secretion of proteases or by the presence of myofibroblasts. ICC staining of the FEB cells on the myofibroblasts marker revealed enhanced expression for αSMA (Figure 3C). 

The morphology analysis revealed that on average, FEB cells are bigger in size, more spread out and less solid in their morphology compared to FHC (Figure 4A). The reason for that is probably the changes in the cytoskeleton associated with traits of cell senescence. It is known that senescent fibroblasts change their morphology from a small spindle-fusiform shape to a large flat broadened shape, and the increase in the surface area could therefore be attributed in part to changes in the cytoskeleton [43]. It has been reported that RDEB patients’ skin is strikingly similar to aged subjects’ skin with its prominent ultrastructural, clinical features and some protein markers [37]. Eukaryotic cells in vivo also steadily increase in size with age. Furthermore, it has been determined that human dermal fibroblasts from aged donors had a different morphology with a larger cellular size [44,45]. Data from our DEG analysis presented below confirm cell aging at some points.

S100A4, or fibroblast-specific protein 1 (FSP1), is considered to be a specific marker of fibroblasts [46]. It is expressed during normal wound healing and is involved in pathological processes as well [47]. Transgelin (TAGL), which is known as a protein specific for fibroblasts and smooth muscle cells, is downregulated during EMT and becomes upregulated in response to TGF-β stimulation [48,49]. ICC analysis of S100A4 and TAGLN expression revealed homogeneous staining across all cells, confirming the fibroblast nature of these cells and the absence of contamination by other cell types (Figure 3A). 

The variety of forms and phenotypic traits in patients with the same diagnosis hinted at the important role of the individual differences that may concern local sequence contexts and differential gene expression (DEG) patterns. The DEG pattern of FEB is affected by the small sample of FEB lines and their heterogeneity due to different COL7A1 mutations. These factors may hamper the delineation of differential expression trends specific to RDEB. Hence, the group of other RDEB fibroblasts’ cell lines was searched for more comprehensive analysis of the DEG pattern. In our research, DEG were studied in RDEB fibroblast lines with the *c.6527insC* mutation, the RNA-seq data of which were available from public resources. 

Firstly, we analyzed the expression of genes related to inflammation or fibrosis. The lack of C7 induces thrombospondin 1 (THBS1)-dependent TGF-β activation, shifting the abundance of ECM content towards the interstitial ECM proteins [7]. Transforming growth factor β (TGF-β) can stimulate or inhibit the processes of tissue homeostasis: it can influence cell proliferation and induce senescence and apoptosis, or induce Epithelial Mesenchymal Transition EMT and promote invasion, activate autophagy or promote metastatic colonization. The consequence of TGF-β activation affects the course of RDEB. It was shown previously that the level of TGF-β1 heavily varies between different RDEB patients [50]. No differences in expression of TGF-β1 were found between FEB and FHC lines or between RDEB and healthy fibroblasts with the c.6527insC mutation. In contrast with this, increased AGTR1 expression was observed in all FEB lines as well as in RDEB transcriptomes with the c.6527insC mutation (Appendix A). Our RT-qPCR data also revealed the diminished level of THBS1 expression in two lines, and the strong modulation of the expression level was observed in all FEB lines (Appendix A). TN-C is considered to be a part of TGF-β cascade activation, and it was clearly elevated only in the FEB2 cell line (Appendix A). 

Secondly, given the fact that RDEB fibroblasts can acquire cancer-associated traits, the expression pattern of S100A4 (FSP1) and TAGL is worth analyzing more specifically by RT-qPCR. We have not observed the enhanced expression of FSP1 in any of the FEB lines (Figure 5A). Taken together with the dwindling expression of the IL1B gene, this could indicate an unchanged mesenchymal state in the FEB lines. 

Along with the FSP1, we investigated the expression of other genes associated with fibroblast reactivation, transdifferentiation into myofibroblasts, or matrix remodeling. We observed different patterns among the FEB lines, with the FEB2 line showing clearly elevated expressions of FAP1 and AOC3 (Figure 5B, AOC3, FAP1). It was previously demonstrated that FAP1 is mainly expressed by the nonmyofibroblast subpopulation of CAFs [51]. AOC3 is not expressed in normal dermal fibroblasts, being the marker of activated fibroblasts [52]. The pronounced expression of both of these genes in the FEB2 line, together with the downregulated expression of TAGL, points out the process of CAF-like transformation. 

Thirdly, we analyzed genes encoding the other components of ECM, proteoglycans, enzymes participating in C7 processing, proteases and matrix-associated protease inhibitors.

All analyzed FEB lines showed the diminished expression of TGM2 (Appendix A). This corresponds with previously published data on the C7-dependent regulation of TGM2 level and activity in RDEB fibroblasts [53]. TGM2 provides cross-linking between C7 fibrils and matrix proteins, so it can be viewed as a multifunctional enzyme involved in different processes. In addition, TGM2 blocks the fusion of autophagosomes and lysosomes. Apparently, TGM2 reduction mediates many RDEB disorders in intracellular and matrix homeostasis. The major substrate of TGM2 is fibronectin, the component of ECM. The alternative splice forms of FN, such as the ED-A + FN form, were shown to be upregulated in specific conditions such as tissue repair or fibrosis [54]. The presence of ED-A + FN activates TGF-β, directing fibrogenesis, and vice versa—the small amount of active TGF-β launches the alternative splicing and synthesis of ED-A + FN. No upregulation of the ED-A + FN transcript was observed in FEB lines (Appendix A). At first glance, this contradicts the observed effect of the enhanced expression of ED-A + FN observed in FEB by ICC (Figure 3B,C). However, the low density of cells used in the ICC experiment may be the reason for the induction of this alternative form of FN that was reported previously [41].

We have shown for the first time the enhanced amount of procollagen c-endopeptidase enhancer 2 (PCOLCE2) mRNA in FEB lines (Figure 5A). This protein participates in the cleavage of procollagen C7 during the secretion and antiparallel dimer formation of mature C7 and was found to compete with procollagen C-proteinase and BMP-1 for the binding to collagen [55]. The enhancement of PCOLCE2 was identified as the distinction between healthy skin fibroblasts and keloid and scar tissue fibroblasts [56]. In addition, PCOLCE2 is recognized as one of the markers of active inflammation [57]. The gene encoding the glycoprotein gliomedin (GLDN) has never been shown among DEG in RDEB. Increased GLDN expression was revealed in all samples except FEB3 (Appendix A, Supplementary Discussion). The enhanced proteoglycan expression indicates the fibrotic changes in the ECM of RDEB cells. The gene encoding the proteoglycan dermatopontin (DPT) was upregulated in all FEB lines except FEB3 (Figure 5A). DPT accelerates and organizes collagen and fibronectin fibril formation and activates TGF-β1. DPT binds decorin (DCN), which may be called its functional antagonist [3]. Despite the fact that DPT facilitates the formation of thinner collagen fibrils, preventing the formation of scars, it can exacerbate the development of fibrosis in pathological cases [58]. Enhanced DCN expression has been associated with milder manifestations of the disease [3]. Downregulation of DCN mRNA in the two lines FEB1 and FEB3 (severe RDEB forms) confirms the reported earlier data (Figure 5A).

The cathepsins are another group of important members in the development of chronic inflammation and matrix damage. Our data demonstrate an increase in CTSZ expression levels in all FEB lines (Figure 5A) and the modulation of expression of cathepsin B (CTSB) in two groups of FEB lines (Figure 5B). CTSZ enhancement fuels autophagic disbalance and ECM remodeling [28]. Therefore, disturbance in ECM remodeling factors may indicate chronic inflammatory conditions, resulting in the increased degradation of structural DEJ proteins and the development of conditions that favor tumor transformation.

One of the most unexpected findings is the diminished level of IL1B mRNA in all FEB samples (Figure 5A). Despite the fact that the sera as well as blister fluid of RDEB patients have elevated levels of this cytokine [58,59], the absence of IL1B upregulation was demonstrated in RDEB with no scarred skin [50]. However, the dynamic of IL1B generally changes in the skin in accordance with the cell metabolism aging effects. Our results confirmed the decrease of IL1B mRNA being common for both RDEB fibroblasts with c.6527insC and our FEB lines.

The important trait of FEB lines is the EREG downregulation (Figure 5B). EREG is a secreted matrisome EGFR ligand capable of reprogramming human normal fibroblasts to cancer-associated fibroblasts (CAF). It also promotes EMT, which is necessary for the migration and invasion of fibroblasts and the progression of oral squamous cell carcinoma [60]. In accordance with our data, EREG expression is upregulated by the IL1B autocrine loop [61]. Furthermore, the expression of EGFR ligands is regulated by integrins [62,63], while the integrin signaling axis is disturbed in RDEB [28]. c.6527insC RDEB transcriptomes did not reveal downregulation of EREG, but they demonstrated the decrease of another matrisome-associated EGFR ligand, AREG. Downregulation of EREG in all FEB lines sheds new light on the state of these RDEB fibroblasts, which have bypassed the possible transition to CAF.

The observed RDEB-associated changes in expression are distinct from those in granulation tissue formation in injured healthy skin. Particularly, in this study, we have demonstrated the upregulation of the genes encoding the glycoproteins GLDN and PCOLCE2 in FEB lines, whereas these genes were found to be among the most downregulated in wound-healing skin. Also, contrary to RDEB, EREG and AREG are upregulated in myofibroblasts in the process of wound healing [64].

Lastly, we searched for genes encoding metalloreductases that participate in inflammation and tumorigenic transformation (Supplementary Discussion). First of all, the upregulation of AOC3 was shown in FEB2 (Figure 5B). Then, we analyzed the expression of the gene encoding the metalloreductase STEAP4. We have shown the enhanced expression of STEAP4 in c.6527insC RDEB transcriptomes. STEAP4 was also upregulated in FEB1, -2, and -4 lines and downregulated in FEB3 (Figure 5B). A distinctive feature of these metalloreductases is the regulation of their expression by proinflammatory cytokines or hormones in response to cellular damage [65]. Thus, the upregulation of STEAP4 in RDEB fibroblasts indicates the activation of cellular defense mechanisms, presumably helping to resist the changes in metabolic function. It may be connected with stable systemic inflammation and high risk of tumorigenic transformation. 

The observed differences in gene expression allows one to discern the FEB3 line from other FEB lines, bringing its expression profile closer to the one of RDEB c.6527insC. As discussed above, the influence of the homozygous missense mutation in the THC domain on the anchoring fiber structure is the reason for such altered patterns of DEG. However, the second heterozygous dominant mutation in FEB3, located in the 2A domain of KRT5 (p.R352C), makes an unknown contribution to this pathological shift of the expression profile due to the additional triggers of inflammation in the skin.

Apart from the separate PANTHER GO BP annotations, we also highlight the GO BP terms that are enriched in both DEGs of the RDEB c.6527insC dataset and DEGs found in our cell lines, as well as the terms that are unique for the DEGs of the FEB lines (Appendix A, Appendix A). GO terms that are associated with the inflammation and fibrosis development were found to be common. Other common GO terms include ‘positive regulation of cellular component movement’, ‘negative regulation of secretion by cell’ and ‘cell surface receptor signaling pathway’. This acts in accordance with the data about dysregulated autophagy and profibrotic changes of ECM in RDEB cells [53]. Among GO BP terms uniquely enriched in FEB lines, we highlight the positive regulation of smooth muscle cell and fibroblast proliferation, negative regulation of cellular response to growth factor stimulus, negative regulation of epithelial cell migration and negative regulation of cellular response to growth factor stimulus.

## 4. Materials and Methods

### 4.1. Obtaining Skin Biopsy and Blood Samples

We recruited 4 subjects with diagnosed RDEB from the patients of the National Medical Research Center for Children’s Health, Moscow. As a control group, the participants with no symptoms or signs of EB were included in the study. The study was conducted in accordance with the Declaration of Helsinki, and approval was obtained from the Local Research Ethics Committee of Pirogov Russian National Medical University. Each patient enrolled in this study provided written informed consent for his or her involvement. Venous blood samples of all subjects were collected in EDTA-containing tubes. Standard technique was used for the isolation of human peripheral blood mononuclear cells (hPBMC), which includes density centrifugation with the Ficoll–Paque gradient as described in [66]. Skin samples were obtained from subjects using a skin biopsy punch (4–6 mm, MEDAX, Poggio Rusco, Italy). All subjects gave informed consent on the publication of the photos.

### 4.2. Immunohistochemistry of Collagen VII in Skin

Unfixed skin was embedded in Tissue-Tek O.C.T. (Sakura, Osaka, Japan) and was frozen in liquid nitrogen vapor. Tissue sections were made by cryotomy. Slides with mounted sections were air-dried and then stored at −70 °C.

Before immunostaining, the sections were fixed using 10% buffered formaldehyde (Biovitrum, Saint Petersburg, Russia). Fixed sections were incubated with primary anti-collagen type VII LH7.2 monoclonal antibodies against NC1 domain (C6805, Merck, Kenilworth, NJ, USA) in block solution based on PBS (PanEco, Moscow, Russia) with 10% fetal bovine serum (Capricorn Scientific, Ebsdorfergrund, Germany) and 0.3% TRITON X-100 (Sigma-Aldrich, St. Louis, MO, USA) at 4 °C overnight and then with secondary antibodies (A-11029, Invitrogen, Carlsbad, CA, USA) in PBS with 0.3% TRITON X-100 for 2 h at room temperature. Nuclei were stained with DAPI (Biotium, Fremont, CA, USA).

Fluorescent microphotographs were obtained using the inverted fluorescent microscope EVOS FL AUTO (ThermoFisher Scientific, Waltham, MA, USA). Image preparation was conducted using Fiji software [67].

### 4.3. Primary Cell Isolation and Culture

Primary fibroblasts and keratinocytes were isolated from skin using a technique described earlier [68]. Briefly, skin was washed in Hank’s Balanced Salt Solution HBSS (PanEco, Moscow, Russia) with gentamicin (0.35 mg/mL, Dalhimpharm, Khabarovsk, Russia). Dermis was detached from the epidermis after incubation with Dispase (Gibco, Grand Island, NY, USA). The detached dermis was then cut into pieces less than 1 mm^3^ and digested by incubation with collagenase type I (Worthington, Lakewood, NJ, USA). Digested dermis was then washed from collagenase and seeded on plastic dishes in a culture media and incubated in a 5% CO_2_ incubator at 37 °C until fibroblast colonies developed.

After incubation of detached epidermis in 0.05% trypsin–EDTA (Gibco, Grand Island, NY, USA), a suspension of cells was obtained by intensive shaking. Cells were then sedimented by centrifugation, resuspended in CnT-07 medium (CELLnTEC, Bern, Switzerland) with addition of Y-27632 (ab120129, Abcam, Cambridge, UK), seeded on culture plastic (Corning, Corning, NY, USA), covered with self-made rat-tail collagen [69] and placed in a 5% CO_2_ incubator at 37 °C. After two passages, keratinocytes were frozen for storage in liquid nitrogen.

The culture medium for fibroblasts was based on DMEM High Glucose (4.5 g/L) (Capricorn Scientific, Ebsdorfergrund, Germany) with 10% fetal bovine serum (Capricorn Scientific, Ebsdorfergrund, Germany), GlutaMAX (Gibco, Grand Island, NY, USA), Sodium pyruvate (Gibco, Grand Island, NY, USA) and PenStrep (Gibco, Grand Island, NY, USA).

Subculture of cells was performed using versene solution (PanEco, Moscow, Russia) and 0.05% trypsin–EDTA (Gibco, Grand Island, NY, USA).

For storage in liquid nitrogen, cells were frozen in the culture medium with 10% DMSO (PanEco, Moscow, Russia). Before placing in liquid nitrogen, cryovials with cells were cooled down from room temperature to −70 °C at a rate of −1 °C/min.

### 4.4. GenBank Accession Number and Clinical Resources

COL7A1 gene (Gene ID: 1294) encodes the alpha chain of type VII collagen. NCBI reference sequence: NG_007065.1. Ensembl: ENSG00000114270 MIM:120120. Location: NC_000003.12. Official full name provided by HGNC: collagen type VII alpha 1 chain. Transcript identifier: LRG_286t. Clinical resources for COL7A1: OMIM 120120, Orphanet 120738, ClinGen Search via HGNC: 2214.

### 4.5. Preparation of DNA, cDNA and Reverse Transcriptase Polymerase Chain Reaction (RT-PCR), One-Tube PCR

Total DNA was isolated from 1 × 10^6^ fibroblasts cells or 5 × 10^6^ hPBMC with Extract DNA Blood & Cells kit (Evrogen, Moscow, Russia) following the manufacturer’s instructions. Total RNA was isolated from 2–4 × 10^6^ cells with ExtractRNA (Evrogen, Moscow, Russia) following the manufacturer’s instructions. Two micrograms of total RNA were treated with DNAse I (ThermoFisher Scientific, Waltham, MA, USA), then after thermal inactivation of DNAse in the presence of EDTA, 1 µg of RNA was used for reverse transcription with a MMLV RT kit (Evrogen, Moscow, Russia), using random (deca-nucleotides) primers for synthesis. RT-PCR was carried out by Encyclo Plus PCR kit (Evrogen, Moscow, Russia) using 1.5 μL of 1:40 dilutions of cDNA template as the templates. Each of the specific primers was used in 10 µM amounts. 

Luna Universal One-Step RT-PCR Kit (New England BioLabs, Ipswich, MA, USA) was used for one-tube RT-PCR with 1 µg of RNA with specific primers (the list of primers is shown in the Appendix A). The thermal cycler C1000 Touch (Bio-Rad, Hercules, CA, USA) was used as the equipment for amplification.

### 4.6. RNA-Seq Analysis

Total RNA was purified using an RNeasy Mini Kit spin column (RNEasy; Qiagen, Hilden, Germany), at room temperature. Library prep was carried out with a NEBNext Ultra II Directional RNA Library Prep kit (New England BioLabs, Ipswich, MA, USA). 

The sequencing platform was an Illumina Hiseq 4000; RNA sequencing was carried out by an external laboratory (Genewiz, Inc; South Plainfield, NJ, USA) and expanded to 10–20 M reads at 2 × 150 bp.

### 4.7. RT-qPCR

RT-qPCR analysis was performed with a RT-PCR SYBR kit (Evrogen, Moscow, Russia) by using 1 μL of 1:40 dilutions of cDNA template. The size of the amplicon was rigorously optimized as well as specific primer pairs in order to prevent primer hybrid amplification in favor of target ampliconproduction. All samples were taken in triplicate. Melting curves were analyzed. We used the ΔΔCT method [70] to analyze relative gene expression with normalization to GAPDH and RPL27 gene expression. Finally, the value U was estimated, which determines the relative gene expression by normalization to the level of GAPDH and RPL27 gene expression. The CFX96 Real-Time System (Bio-Rad, Hercules, CA, USA) thermal cycler was used.

### 4.8. Genetic Constructs

Purification of PCR products and products of digestion was performed by gel electrophoresis and subsequent extraction with the Cleanup Standard Kit (Evrogen, Moscow, Russia). The cDNA fragments of COL7A1 from FEB cells were cloned into the pAL-TA (Evrogen, Moscow, Russia), and the plasmids were used for Sanger sequencing (Evrogen, Moscow, Russia).

### 4.9. Cell Morphology Analysis

Fibroblasts were seeded at low density so the single cells could be separated from each other on microscopic images. Phase-contrast photographs of the cells were made using EVOS FL AUTO the day after seeding. The PHANTAST FiJi plugin [71] was used to segment cells on the images. Holes in binary masks were filled and shapes were analyzed with the standard FiJi function ‘Analyse particles’. The following parameters were analyzed: area, perimeter, bounding rectangle (width, height and aspect ratio), fit ellipse (major and minor axes), circularity, roundness, solidity and caliper diameter (max and min). The estimation of cell size and granularity was made by analyzing forward (FSC) and side (SSC) scatter from flow cytometry data from a Bio-Rad S3 Cell Sorter (Bio-Rad, Hercules, CA, USA).

### 4.10. Immunocytochemical Fluorescence Analysis

Cells were fixed using 10% buffered formaldehyde (Biovitrum, Saint Petersburg, Russia). The fixed cells were incubated with primary antibodies in a block solution based on PBS (PanEco, Moscow, Russia) with 10% fetal bovine serum (Capricorn Scientific, Ebsdorfergrund, Germany) and 0.3% TRITON X-100 (Sigma-Aldrich, St. Louis, MO, USA) at 4 °C overnight and then with secondary antibodies in PBS with 0.3% TRITON X-100 for 2 h at room temperature. Nuclei were stained with DAPI (Biotium, Fremont, CA, USA).

The antibodies used were:-Primary anti-collagen I antibodies (RAH C11, Imtek, Moscow, Russia)-Primary anti-collagen IV antibodies (ab6586, Abcam, Cambridge, UK)-Primary anti-fibronectin antibodies (ab2413, Abcam, Cambridge, UK)-Primary anti-S100A4 antibodies (ab27957, Abcam, Cambridge, UK)-Primary anti-α-SMA antibodies (ab5694, Abcam, Cambridge, UK)-Primary anti-FN ED-A antibodies (ab6328, Abcam, Cambridge, UK)-Primary anti-SM22ɑ antibodies (ab10135, Abcam, Cambridge, UK)-Primary anti-collagen VII antibodies (C6805, Merck, Kenilworth, NJ, USA)-Secondary anti-mouse Alexa-594 (A21201, Invitrogen, Carlsbad, CA, USA)-Secondary anti-mouse Alexa-488 (A11029, Invitrogen, Carlsbad, CA, USA)-Secondary anti-rabbit Alexa-594 (A21442, Invitrogen, Carlsbad, CA, USA)-Secondary anti-goat Alexa-488 (A21467, Invitrogen, Carlsbad, CA, USA)

### 4.11. Confocal Imaging

Confocal images were made using a LSM 880 confocal scanning microscope (Carl Zeiss Microscopy GmbH, Jena, Germany) based on an inverted fluorescent microscope Axio Observer.Z1 Zeiss equipped with six laser lines (633, 594, 561, 543, 514, 488 and 405 nm), five objectives (EC Plan-Neofluar 5×/0.16, EC Plan-Neofluar 10×/0.3, PL APO 20×/0.8, PL APO 40×/0.95, PL APO 63×/1,4 Oil DIC) and LSM-software ZEN 2. The following emission bands were used: DAPI, 410–579 nm; FITC, 493–579 nm; TurboFP635, 582–754 nm; Alexa Fluor-594, 585–733 nm.

### 4.12. Semiquantitative Immunocytochemical Collagen VII, α-SMA and FN ED-A Expression Assay

Fibroblasts were seeded into the wells of 96-well plates (Corning, Corning, NY, USA) in amounts of 6 × 10^3^ cells/well. The cells were cultured for 7 days and then fixed with 10% buffered formaldehyde and stained with primary anti-collagen VII antibodies (ab93350, Abcam, Cambridge, UK) and secondary anti-rabbit antibodies conjugated with Alexa-594 (A21442, Invitrogen, Carlsbad, CA, USA). Nuclei were stained with DAPI (Biotium, Fremont, CA, USA). Fluorescent microphotographs of the wells were made with the EVOS FL AUTO microscope with the same channel settings for all images. At least 3 fluorescent microphotographs were made for each fibroblast line.

The threshold function of Fiji software [67] was used to select stained cells on the images. Staining intensity of the cells was evaluated as the mean pixel intensity value of the thresholded area for every image.

The same method was applied to evaluate α-SMA and FN ED-A expression. Primary anti-ɑSMA antibodies (ab5694, Abcam, Cambridge, UK) and anti-FN ED-A antibodies (ab6328, Abcam, Cambridge, UK) were used.

### 4.13. Collagen Gel Contraction

Fibroblasts were embedded into a gel made of rat-tail collagen type I [69] with collagen concentration of 3 mg/mL and fibroblast concentration of 1 × 10^5^ cells/mL. Into each well of 24-well plates was placed 500 μL of liquid gel with cells. After gelation, 500 μL of culture media were added into each well. The following day, gels were detached from the walls of the wells using syringe needles and gels were allowed to contract. A scan of each of them was made using the EVOS FL AUTO microscope on day 2, and the area of each collagen tablet was measured using Fiji software [67]. 

### 4.14. Western Blot Analysis

Cell were lysed in denaturing lysis buffer (RIPA), diluted in Laemmli buffer 1:1 and loaded into SDS-PAGE gel: 8% gel with 8 M urea. Then, separated proteins were transferred to the nitrocellulose membrane. Before the semidry transfer, a part of the gel containing C7 was washed in a series of up to 3% acetic acid solutions. Then, the transfer was conducted in a 3% acetic acid solution, the NC membrane was closer to the cathode and the gel was closer to the anode during the transfer. Membranes were blocked with 3% (*w/v*) nonfat dry milk in TBST for 1 h, incubated overnight at 4 °C with the corresponding antibody solutions: anti-collagen VII, anti-β-actin. Then, they were washed, incubated with the corresponding HRP-conjugated secondary antibody and imaged by an electro-chemiluminescence (ECL) kit and imaging system. The detailed list of reagents and equipment and protocols of lysates’ preparation, PAAG and transfer procedures are in the Appendix A (WB of type VII collagen).

### 4.15. RNA-Seq Data Analysis

Raw FASTQ files for RNA-seq data of nine RDEB patients and three healthy controls were downloaded from the Gene Expression Omnibus under the accession number GSE119501. The included patients were 50% female and ranged from 13 days to 42 years old. The sequencing platform of this dataset was a GPL1154 Illumina Hiseq 2000. Sequences were trimmed using cutadapt software (v.2.8) with the parameters -q 30 -m 50 -a AGATCGGAAGAGC [72]. The quality of the data was accessed using FastQC (v0.11.8) software [18]. Trimmed reads were aligned to the human reference genome (GRCh38) using Hisat2 (v.2.1.0) [73] with the parameters no-softclip—dta and genome_snp_tran index file. Read count matrix was calculated by Stringtie (v2.0.6) [74] software with parameters -B -e and read into EdgeR software (v.3.30.3) [75].

We used the classic EdgeR pipeline for the downstream analysis. Briefly, after filtering out lowly expressed genes, effective library sizes were computed by the trimmed mean of the M-values normalization method [76]. Counts per million (CPMs) were extracted based on the normalized expression values. Differential expression between healthy and RDEB samples was calculated using a likelihood ratio test after fitting the generalized linear model.

### 4.16. Gene Ontology Analysis

For the RNA-seq data, DEGs with Benjamini–Hochberg false discovery rate (FDR) <0.05 and log2 fold change (logFC) >1.5 and <1 were selected for gene ontology (GO) analysis. These genes were annotated by the overrepresentation test against the PANTHER GO complete biological process database. A similar procedure was performed on DEGs found by the RT-qPCR, and GO terms common between the two datasets were visualized by the R package ‘rrvgo’ [77], which groups the GO terms according to semantic similarity and enrichment score. The enrichment score from RT-qPCR-derived PANTHER analysis was used for common terms.

### 4.17. Software

Statistical calculations were made using Microsoft Excel, Origin and GraphPad Prism.

Image editing and analysis were performed using PaintNET, Adobe Photoshop, InkScape and Fiji.

Flow cytometry files were analyzed using FlowJo.

The quality of raw fastq reads was assessed by FastQC.

The alignment of fastq reads to the reference genome was done by Hisat2.

RNA-seq feature counting was performed by Stringtie.

Normalization and differential expression of RNA-seq read counts was performed by EdgeR.

## 5. Conclusions

A panel of specific cell lines from human dermal fibroblasts was established, with four FEB lines from RDEB patients being among them, as well as a group of FHC lines. The presence of recurrent mutations in the patients’ COL7A1 gene was confirmed. The impairment of COL7A1 splicing in three FEB lines was found, and the presence of mRNA of COL7A1 for all FEB lines was confirmed. Several obtained properties of the FEB cell lines, namely their morphology and contraction in the collagen gel assay, were compared with healthy control cell lines. Flow cytometry showed that FEB cells were less circular and solid and larger on average than FHC cells. FEB cells were more active in modifying collagenous matrices than FHC cells. ICC staining of the FEB cells’ myofibroblast markers revealed the enhanced expression for αSMA and the FN ED-A form of fibronectin. In order to compare the patterns of RDEB mutation effects on cellular homeostasis, we analyzed the available RDEB fibroblasts’ RNA-seq data with the c.6527insC mutation. Several differentially expressed genes were shown to be common for the FEB lines. The dwindling expression of DSP and IL1B are among them, together with the elevated expression level of several factors known to be associated with ECM remodeling. Enhanced expression of DCN and PPARG correlated with milder forms of RDEB. The gene signatures of FEB lines drive them to remodel the ECM and acquire the traits of CAFs. The balance of profibrotic and antifibrotic factors depends on mutation type, a fact evident in the comparison of RDEB fibroblast lines with varying COL7A1 transcript integrity.

## Figures and Tables

**Figure 1 ijms-22-01792-f001:**
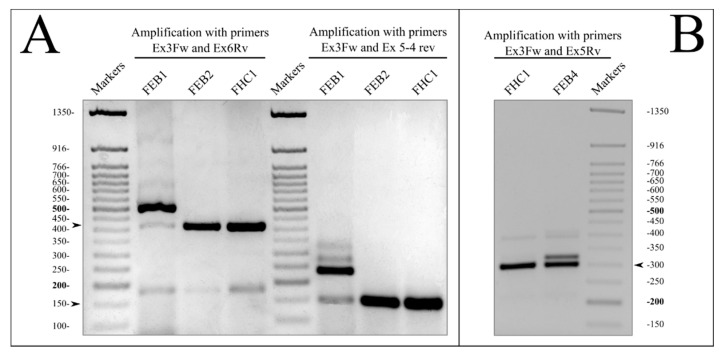
RT-PCR analysis of COL7A1. Markers-Fast DNA Ladder 50 bp (New England BioLabs, Ipswich, MA, USA). (**A**). COL7A1 splicing in the FEB1, FEB2 and FHC1 lines. PCR with two pairs of primers, corresponding to exon 3 and exon 6 (Ex3fw, Ex6Rv) and exon 3 and the exon 5-exon 4 junction (Ex3fw, Ex5-4rv), respectively. Normal splicing forms, indicated by arrows, represent the bands of 420 bp and 150 bp in lanes FEB2 and FHC1. For FEB1, the presence of the normal splicing form is clearly visible. The major splice form of FEB1 is the intron-retaining form with length of 505 bp. (**B**). COL7A1 splicing in FHC1 and FEB4. PCR with exon 3- and exon 5-specific primers (Ex3fw, Ex5Rv). Normal splicing forms (indicated by the arrows) represent the band of 300 bp; the aberrant splicing form represents the band of 320 bp in FEB4 lane. For uncut images, see Appendix A.

**Figure 2 ijms-22-01792-f002:**
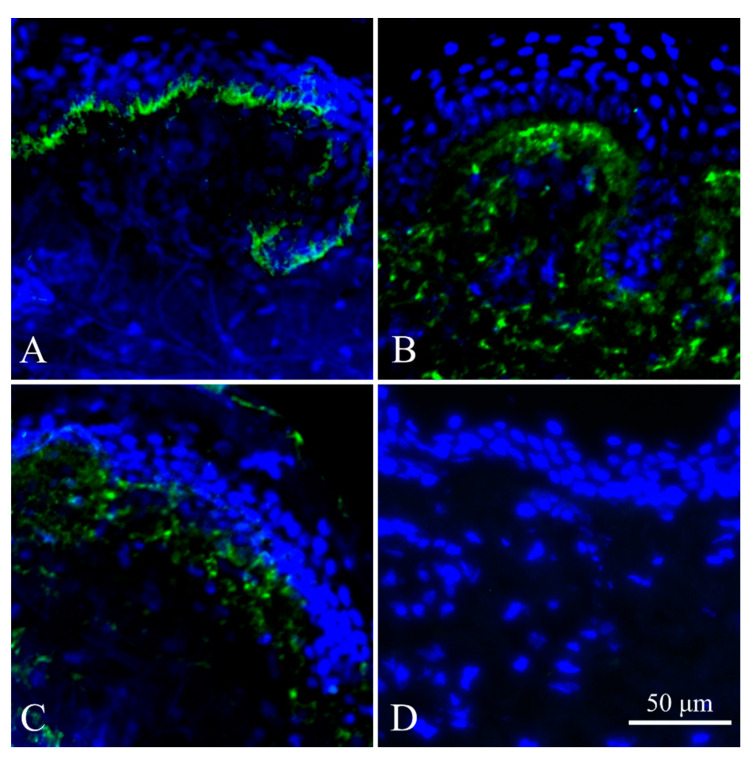
Immunohystochemistry IHC staining of skin cryosections. (**A**) Skin of healthy donor d6. (**B**) Skin of patient d2 with RDEB inversa. (**C**) Skin of patient d3 with generalized severe RDEB. (**D**) Autoimmunofluorescence of skin. Green channel—type VII collagen. Blue channel—DNA (DAPI). Fluorescent microscopy.

**Figure 3 ijms-22-01792-f003:**
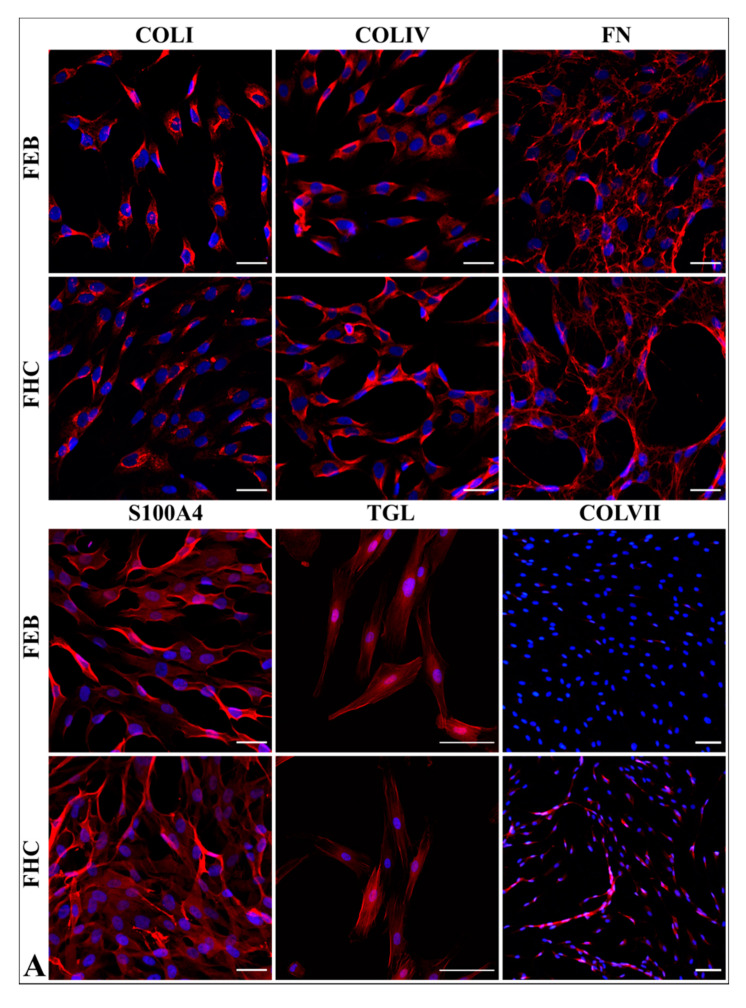
Immunocytochemistry ICC staining of FEB1 (FEB) and FHC1 (FHC). (**A**). Red channel shows markers: COLI—type I collagen; COLIV—type IV collagen; FN—fibronectin; S100A4—fibroblast-specific protein 1; TGL—transgelin; COLVII—type VII collagen. Blue channel shows nuclei (DAPI). Scale bar—100 µm. Confocal microscopy (COLI, COLIV, FN, S100A4, TGL) and fluorescence microscopy (COLVII). (**B)**. Red channel shows ɑSMA—alpha smooth muscle actin. Green channel shows FN ED-A—the ED-A segment of fibronectin. Blue channel shows nuclei (DAPI). Scale bar—100 µm. Confocal microscopy. (**C**). ɑSMA expression level in FEB and FHC lines. Mean value and 95% confidence interval shown for combined FEB (1–4) and FHC1 (FHC). The FHC was taken as 100%. (**D**). FN ED-A expression level in FEB and FHC lines. Mean value and 95% confidence interval shown for combined FEB (1–4) and FHC1 (FHC). The FHC was taken as 100%.

**Figure 4 ijms-22-01792-f004:**
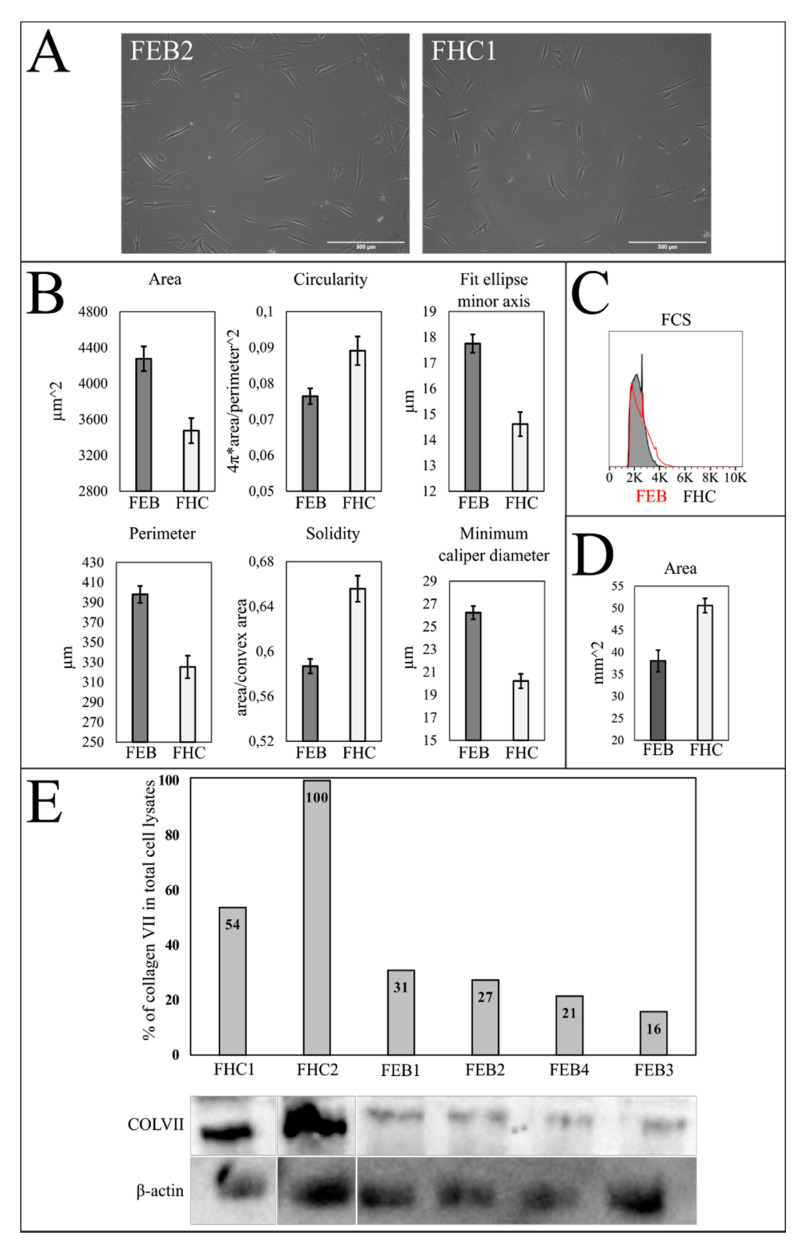
FEB and FHC comparison. (**A**) Phase-contrast microscopy of fibroblasts in culture. Magnification ×100. (**B**) The results of morphological analysis of FEB and FHC. Area, perimeter, minimum caliper diameter and fit ellipse minor axis are significantly bigger (*p* < 0.05) and circularity and solidity are significantly lower (*p* < 0.05) for FEB compared to FHC when compared by two-sample *t* test. Error—standard error. (**C**) Forward Scatter FSC histogram of FHC and combined FEBs (1–4). Distributions are significantly different (T(x) = 868; *p* < 0.05). (**D**) Collagen gel contraction assay. The results of the Mann–Whitney test show that distributions are significantly different (*p* < 0.05). (**E**) Western blot analysis of total cell lysates. Semi-quantitative estimation of type VII collagen expression. β-actin used as a loading control. SDS-PAGE (8%) with 8 M urea, anti-collagen VII polyclonal antibody (upper panel) and anti-β-actin (lower panel); electro-chemiluminescence (ECL) detection. For the uncut and unadjusted version, see Appendix A.

**Figure 5 ijms-22-01792-f005:**
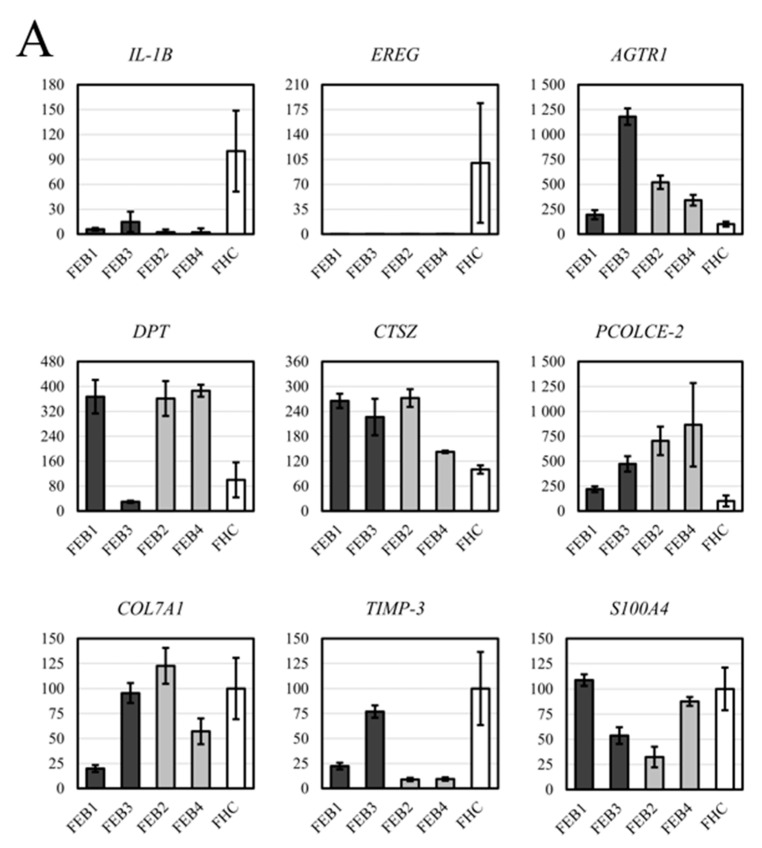
RT-qPCR analysis of DEG in FEB and FHC lines. Dark grey and light grey column colors indicate cell lines from patients with severe (FEB1, FEB3) and mild (FEB2, FEB4) forms of EB, respectively. White column color indicates the healthy control. The ordinate value is U—relative gene expression, estimated as ΔΔCT, normalized to the mean of FHC. The mean of FHC was taken as 100%. The mean value and 95% confidence level is given for each FEB line and the average value of pooled data from several FHC lines. (**A**). Strong downregulation of *IL1B, EREG* genes was demonstrated, the upregulation of *AGTR, PCOLCE-2,CTSZ* were shown. (**B**). *PPARG*, *CXCR4*, *DCN*, *MELTF*, *IL7* and *CTSB* were increased in the milder group and decreased in the severe FEB group. *DSP* and *TIMP3* were downregulated in all FEB lines, with a tendency to be further decreased in mild groups of FEB.

**Table 1 ijms-22-01792-t001:** List of patients with the recessive form of dystrophic epidermolysis bullosa (RDEB) and healthy control donors: gender and age information, numbers of patient-specific cell lines obtained and mutations identified in the *COL7A1* gene.

Patient	Form of RDEB	Gender/Age	Cell Line	C7 Level	Mutation 1	Mutation 2
d1 RDEB	generalized severe ORPHA 79408	M/8	FEB1	diminished	*c.425A > G* *(p.K142R)*	*c.425A > G* *(p.K142R)*
d2 RDEB	inversa ORPHA 79409	M/16	FEB2	diminished	*c.682 + 1G > A*	*c.6205 C > T* *(p.R2069C)*
d3 * RDEB	generalized severe ORPHA 79408	M/5	FEB3	diminished	*c.8245G > A* *(p.G2749R)*	*c.8245G > A* *(p.G2749R)*
d4 RDEB	generalized intermediate ORPHA 89842	M/21	FEB4	diminished	c.520G > A (p.G174R)	undetermined
d5	-	F/58	FHC1	normal		
d6	-	F/57	FHC2	normal		
d7	-	M/8	FHC3	normal		
d8	-	F/24	FHC4	normal		
d9	-	M/24	FHC5	not shown		
d10	-	F/25	FHC6	not shown		
d11	-	M/35	FHC7	not shown		
d12	-	M/8	KHC8 **	not shown		

*—Proband had a heterozygous mutation *c.1054C > T* (*p.R352C*) in the keratin 5 (*KRT5*) gene. **—Keratinocytes of healthy control donor. FEB: RDEB fibroblast lines; FHC: control fibroblasts from healthy donors.

**Table 2 ijms-22-01792-t002:** Morphology analysis of individual FEB lines compared to FHC1.

Cell Line	Area	Perimeter	Fit Ellipse Minor Axis	Circularity	Solidity	Minimum Caliper Diameter	Contraction of Collagen Gel	FSC	SSC
FEB1		high		low	low				high
FEB2	high	high	high		low	high		high	high
FEB3					low		high	low	low
FEB4	high	high			low	high		high	high

Statistics done by using the Fisher LSD test and one-factorial ANOVA. Contraction of collagen gel by individual FEB lines was compared to FHC1 using the Mann–Whitney test. FSC and SSC of FEB were compared to FHC1 by the chi-squared comparison. ‘High’ and ‘low’ indicate that a FEB line has a larger or smaller value, respectively, compared to FHC, and the difference is significant (*p* < 0.05).

## Data Availability

Not applicable.

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
