# Peer review of "Signatures of Dermal Fibroblasts from RDEB Pediatric Patients"

_ijms, 2021, doi:10.3390/ijms22041792_

Round 1

Reviewer 1 Report

After reviewed the manuscript of Beilin et al “Signatures of dermal fibroblasts from RDEB pediatric patients”, I have not mayor critic toward the article, but I am sorry to say that no novelty in the work is showed. Several articles are right now publishing (and as is cited in this article for the authors) with more exhaustive methodology focusing, not only, in the blistering problem, but also in all the devastating effects that this entails, such as the great inflammation and fibrosis that ultimately leads to the appearance of SCCs in these patients at an early age, being the main cause of death in these patients. This article sum 4 more patients to the already findings data, that of course, enhance the knowledge of this complicate disease to the scientific community and help , as is mentioned for the authors in the article  : “The model characterizing the stages of fibrogenesis could also be effective for the evaluation of the new  therapeutic inhibitors of the fibrotic cascade”. (line  74-75). So, I suggest additional experiments to bring some novelty to the work.

  1. Although the authors describe the different mutations and theirs characteristics over the different sections, I think will be more clear for the reader expand Table 1 with all  this information as well ( Ej: subtype of DEB, expression or not of COL7……)
  2. In section 2.3: Immunohistochemistry of skin biopsy (IHC). Authors cite: “Skin biopsies from d6 and d3 were studied for C7 expression (Figure 1). Whereas in  the healthy skin C7 is located around the dermoepidermal junction, in the skin of the  RDEB patient the expression of C7 seems to be lower and fails to relocate to the  sublamina densa, instead spreading across the dermis in the skin” ( line123-127)

I would like authors show a negative control of the IF to be sure that the staining in b is not background.

  1. In section 2.4.1: Fibroblasts markers expression. Authors cite: “Fibroblasts from both FEB and FHC lines were analyzed for the expression of fibroblasts markers: S100A4, collagen IV, collagen I, fibronectin and type VII collagen (Figure 2). All cells showed the equal distribution of markers with the single exception of C7 in FHC and FEB lines, the latter having a vanishingly low C7 level. We applied the method of semi-quantitative comparison of ICC data of the C7 expression level. Staining intensity for C7 was measured for FEB (1-3) and FHC (1-4). The means of these groups were compared by nested t test. The results show that FEB cells have lower C7 staining intensity compared to FHC cells ( p<0,001) (Supplementary Figure S2; table S1)” (Lines 137-144)

Authors study expression of S100A4, as fibroblast marker (line484-486). It is published that FSP1 is expressed in normal tissue during normal wound healing or different disease. But In addition to this role, FSP1 has also been implicated in cancer progression, especially among tumor-associated stromal cells; suggest that S100A4 is a crucial regulator of CAF development in the tumor stroma. (Bussard KM, Mutkus L, Stumpf K, Gomez-Manzano C, Marini FC. Breast Cancer Res. 2016 Aug 11;18(1):84.  The result of the authors of similar expression of this protein in both FEB and FHC I think required more attention.

  1. In section 2.4.3: Contraction of collagen gel, authors cite: “The contraction of four collagen gels was measured for each FEB (1, 2, 3 and 4) and FHC1. The final data was combined into two groups (FEB and FHC) and compared by Mann-Whitney test. The results show that fibroblast-induced collagen contraction was significantly higher in FEB cells in comparison with FHC cell lines (p<0,05). The smaller the area of the gel, the higher the cell contraction recorded (Figure 3D; Table 2). (lines 181-186)

Authors perform collagen contraction gel as Functional assay. However, the difference must be interpreted mechanistically as well. Therefore I would suggest that some attention must be dedicated to the quantification of myofibroblasts in studied fibroblasts (immunocytochemistry for this purpose would be consider).

  1. In section 2.6: Transcriptomes analysis. Authors  described: To investigate the key factors of disease progression, we analyzed RNA-seq data  from the study of RDEB fibroblasts from patients of different age and sex, but have the  same mutation c.6527insC”. (line 255-257).

That is not clear for me if RNA-seq analysis are done in patients with c.6527insC  mutation or in their cell line( as is mentioned below)

  1. In section:2.7: RT-qPCR analysis for DEG of FEB and FHC, Authors cite: “To evaluate the differential expression  of fibronectin (FN1), the alternatively spliced form with ED-A+ exon (FN-209) was  selected for RT-qPCR analysis. It was slightly downregulated in FEB samples  (Supplementary Figure S7).( line 302-304)

Nevertheless in the IF ( figure2), the expression of FN is similar as you mentions:  “Fibroblasts from both FEB and FHC lines were analyzed for the expression of fibroblasts  markers: S100A4, collagen IV, collagen I, fibronectin and type VII collagen (Figure 2).”( line 137). Although it is true authors discuss this phenomena in the discussion: “Collagen I, collagen IV and fibronectin are the parts of anchoring fibrils and can  bind to NC-1 domains of C7 [51]. We suggested that defects in C7 in RDEB fibroblasts could cause changes in expression patterns of these proteins. We have performed  semiquantitative analysis of C7 expression using polyclonal antibodies and showed significant reduction in staining intensity for all FEB lines (Figure 2, Supplementary  Figure S2, Supplementary Table S1), but when staining the same cells against collagen I,  collagen IV and fibronectin we have found no visually distinguishable differences in 494 expression patterns for markers investigated between FEB and FHC (Figure 2).” ( line488-493)

This clarification is not clear for me. I think Supplementary Figure S2 need to be better explained

Author Response

We are very grateful to the Reviewer for all critical comments because the suggested additional experiments added depth and value to the work. We hope that our results will be useful to the scientific community, especially for those who are working with the inherited epidermolysis bullosa and other human disorders that involve profibrotic changes and CAF formation.

Point 1: Although the authors  describe the different mutations and theirs characteristics over the different sections,  I think will be more clear for the reader expand Table 1 with all  this information as well (Ej: subtype of DEB, expression or not of COL7……)

Response 1: We expanded the Table 1 with additional information, highlighted by yellow.

Point 2: I would like authors show a negative control of the IF to be sure that the staining in b is not background.

Response 2: After the revision, we added the negative control of immunostaining to the Figure, which was renamed from Figure 1 to Figure 2 (Figure 2, D). On this Figure we also added the skin immunostaining of patient d2, Figure 2B. We also added lines: «The negative control of immunostaining demonstrated that ICH test is specific (Figure 2, D)».

Point 3: Authors study expression of S100A4, as fibroblast marker (line484-486). It is published that FSP1 is expressed in normal tissue during normal wound healing or different disease. But In addition to this role, FSP1 has also been implicated in cancer progression, especially among tumor-associated stromal cells; suggest that S100A4 is a crucial regulator of CAF development in the tumor stroma. (Bussard KM, Mutkus L, Stumpf K, Gomez-Manzano C, Marini FC. Breast Cancer Res. 2016 Aug 11;18(1):84.  The result of the authors of similar expression of this protein in both FEB and FHC I think required more attention.

Response 3: We are grateful to the Reviewer for the critical comment and the suggestion to pay attention on the expression of S100A4 marker.

  • We have performed RT-qPCR analysis of FEB and FHC cells, showing that that S100A4 (FSP1) is not upregulated in FEB lines (Figure 5A). In the Results, chapter 2.7, we added:

«S100A4 expression did not differ much from the control lines in FEB1 and FEB3, but was decreased significantly in FEB2 and FEB4 (Figure 5A).»

In the Discussion, we added:

«S100A4 or Fibroblast-specific protein 1 (FSP1) is considered to be a specific marker of fibroblasts [48]. It is expressed during normal wound healing and is involved in the pathological processes as well [49]. Smooth muscle protein 22-alpha or Transgelin (TAGL) is known as a protein specific for fibroblasts and smooth muscle cells, is downregulated during EMT and becomes upregulated in response to TGF-β stimulation [50,51]. ICC analysis of S100A4 and TAGLN expression revealed homogeneous staining across all cells, confirming fibroblasts' nature of these cells and the absence of contamination by other cell types (Figure 3A).»

And below, Section 3.2 of the Discussion, DEG:

«Given the fact that RDEB fibroblasts can acquire cancer associated traits, it is worth to analyse the expression pattern of S100A4 (FSP1) and TAGL more specifically by RT-qPCR. We have not observed the enhanced expression of gene encoding FSP1 in any of FEB lines (Figure 5A). Taken together with the dwindling expression of IL1B gene, this could indicate an unchanged mesenchymal state in FEB lines.» 

  • We were encouraged to perform an additional RT-qPCR to test the expression of two other genes (FAP1, AOC3) which were reported to be specific for CAF or activated fibroblasts. The results are shown on the Figure 5B.

Point 4: Authors perform collagen contraction gel as Functional assay. However, the difference must be interpreted mechanistically as well. Therefore I would suggest that some attention must be dedicated to the quantification of myofibroblasts in studied fibroblasts (immunocytochemistry for this purpose would be consider).

Response 4: We wish to thank the Reviewer for this suggestion because it allowed us to better understand the processes taking place in the investigated cell lines. We performed immunostaining for myofibroblasts marker É‘-SMA and revealed the enhanced expression of this protein in FEB lines. We added another Figure (Figure 3B) with ICH images and the results of semi-quantitative analysis of expression and added the following chapter to the main text: 2.5.1. Fibroblasts markers expression. We added the following lines to  Chapter "3.1. RDEB patients, COL7A1 mutations":

 «We tested whether the enhanced contraction of FEB cells may be explained by the heightened secretion of proteases or by the presence of myofibroblasts. ICC staining of the FEB cells on the myofibroblasts marker revealed enhanced expression for α-SMA (Figure 3B,B).»

Figure 3B contains the respective figure legends.” ICC staining of FEB1 (FEB) and FHC1(FHC). Red channel - É‘SMA, alpha-smooth muscle Actin. … B. É‘SMA expression level in FEB and FHC lines. Mean value and 95% confidence interval showed for combined FEB (1-4) and FHC1 (FHC). The FHC was taken as 100%.”

We also performed ICH on Transgelin (TAGL) and added the result to Figure 3A, and in the main text  mentioned above:

«Transgelin (TAGL) which  is known as a protein specific for fibroblasts and smooth muscle cells, is downregulated during EMT and becomes upregulated in response to TGF-β stimulation ..... ICC analysis of ... TAGLN expression revealed homogeneous staining across all cells...»

We performed also RT-qPCR on TAGL, the results are displayed on Supplementary Figure7S, and we added to the main text, Section 2.7 of the Results:

«..... Interestingly, only FEB4 line has the increased expression of both Alpha-smooth muscle actin (α-SMA) and Transgelin (TAGL) myofibroblasts markers, while FEB1 showed downregulation of these genes (Supplementary Figure S7).»

Point 5: That is not clear for me if RNA-seq analysis are done in patients with c.6527insC  mutation or in their cell line( as is mentioned below)

Response 5: RNA-seq analysis was performed on the fibroblasts cell lines originating from patient’s skin biopsies.

Point 6. In section:2.7:  RT-qPCR analysis for DEG of FEB and FHC,  Authors cite: “To evaluate the differential expression  of fibronectin (FN1), the alternatively spliced form with ED-A+ exon (FN-209) was  selected for RT-qPCR analysis. It was slightly downregulated in FEB samples  (Supplementary Figure S7).( line 302-304)

Nevertheless in the IF ( figure2), the expression of FN is similar as you mentions: 

Response 6: We wish to thank the Reviewer for this comment, as it motivated us to perform the ICH study with antibodies specific to FN-ED-A form (Figure 3B, A). The FEB cells were double immunostained against É‘-SMA and FN-ED-A markers. The results of the semiquantitative estimation is shown on Figure 3B, C. FEB cells have enhanced level of FN-EDA expression. We added the following lines in the main text of Discussion, 3.1. RDEB patients, COL7A1 mutations:

«Semi-quantitative analysis showed the increased expression of alternative form of fibronectin - FN ED+A (Figure 3B, C), but no difference in total fibronectin expression. This difference was observed in low density seeded fibroblasts (Figure 3B, A) but the situation in a dense population could be different [42].»

We would like to note that despite the slight downregulation of FN-ED+A mRNA level in FEB cells, we observed the significant upregulation on the protein level. The low density of cells used in the ICC experiment may be the reason for the induction of this alternative form of FN, that was reported previously (reference [42] in the manuscript). In these conditions, FEB cells were significantly different from the FHC on the level of FN-ED+A form of protein.

Point 7: This clarification is not clear for me. I think Supplementary Figure S2 need to be better explained

Response 7: We made changes in the Chapter 4 of the “Material and Methods” and elaborated more on the used method.

«4.11. Semi-quantitative immunocytochemical collagen VII, É‘-SMA and FN ED-A expression assay

Fibroblasts were seeded into the wells of 96 well plates (Corning, Corning, New York, United States) in amounts of 6*103 cells/well. The cells were cultured for 7 days and then fixed with 10% buffered formaldehyde and stained with primary anti-collagen VII antibodies (ab93350, Abcam, Cambridge, UK) and secondary anti-rabbit antibodies conjugated with Alexa-594 (A21442, Invitrogen, Carlsbad, California, USA). Nuclei were stained with DAPI (Biotium, Fremont, California, USA). Fluorescent microphotographs of the wells were made with the EVOS FL AUTO microscope with the same channel settings for all images. At least 3 fluorescent microphotographs were made for each fibroblast line.

Threshold function of FiJi software [75] was used to select stained cells on the images. Staining intensity of the cells was evaluated as the mean pixel intensity value of the thresholded area for every image.

The same method was applied to evaluate É‘-SMA and FN ED-A expression. Primary anti-É‘SMA antibodies (ab5694, Abcam, Cambridge, UK) and anti-FN ED-A antibodies (ab6328, Abcam, Cambridge, UK) were used.»

Reviewer 2 Report

Over the last 30 years, we have understood most of the mechanisms leading
to monogenic disorders. What we now need is not identifying the variant in the DNA sequence, but being able to interpret their functional effect in the patient. 

Beilin and co-workers created primary patient-specific RDEB fibroblasts lines and compared it with control fibroblasts from healthy donors. They investigated the morphological features and the contraction capacity of the cells. They performed RT-qPCR gene expression analysis of the cell lines (based on the available RNA-seq data) and elucidated mechanisms involved in disease progression.

Very impressive work!

Author Response

We wish to thank the Reviewer for the attention to our work. We agree completely with the Reviewer's comment about the necessity of “being able to interpret their functional effect (i.e. mutations)  in the patient”. These are the driving forces of the current investigation.

Reviewer 3 Report

The study included a low number of patients. Discussion is extremely long, and it is about nothing. In addition, the authors evaluated signatures of dermal fibroblasts from RDEB Russian pediatric patients but in the transcriptome analysis they used Spanish RDEB 251 population. The authors forgot about the personal and population diversity. They should perform a microarray analysis on their own group. IHC stainig analysis was performed only 2 patients. The study needs a really major revision.

Author Response

We wish to thank the Reviewer for the critical comments, as they helped us to improve both the study and the data presentation of our study.

Point 1: “The study included a low number of patients.”

Response 1: Thank you for this comment, we completely agree. However, it should be mentioned that no Russian General Register for RDEB exists yet. Therefore, the search for such patients is coupled with serious difficulties. It also means that a large human population remains undescribed in terms of prevalent forms of RDEB, recurrent disease-causing mutation variants, the spatial distribution of families with RDEB and the degree of random inbreeding between them. The cell lines from the patients we managed to enroll in the study are the first which were characterized in a way we have chosen. We hope that this work will drive the scientific community to create the collection of such cell lines and generate more data about RDEB in Europe and the world at large.

We remodeled the Discussion which was indeed quite long, optimizing the division of chapters, omitting insignificant details, and trying to be more concise in the interpretation of our data.

Point 2: authors evaluated signatures of dermal Fb from RDDEB Russian patients but in transcriptomes analyses they used Spanish RDEB data”, “authors forgot about personal and population diversity”.

Response 2: Firstly, we would like to thank the Reviewer for raising this important issue. We agree that the transcriptome of each individual bears the traits which reflect the individual and populational variability. At the same time, the general trend in RDEB molecular studies is to reveal the gene expression patterns inherent to RDEB irrespective of particular genetic variant. We have made the following comments concerning the inclusion of Spanish transcriptomes in our manuscript.

We added the following fragment to the Section 2.6 of the Results:

To find the common and specific traits of DEG patterns in RDEB, we focused on the RNA-seq data of other fibroblasts induced by c.6527insC mutation

Bearing in mind the small size of the sample, the heterogeneity of FEBs with respect to COL7A1 mutations, and their possible different genetic backgrounds, the group of other RDEB fibroblasts cell lines with another recurrent mutation needs to be added for the more comprehensive DEG analysis. That is why we decided to use publicly available data from RDEB fibroblasts lines with c.6527insC mutation.

We added the following fragment to the Section 3.4 of Discussion:

The DEG pattern of FEB is affected by the small sample of FEB lines and their heterogeneity due to different COL7A1 mutations. These factors may hamper the delineating of RDEB differential expression specific trends. It was the reason that the group of other RDEB fibroblasts cell lines with recurrent mutation was searched for more comprehensive analysis of DEG pattern specific for RDEB. In our research, DEG were studied in RDEB fibroblasts lines with c.6527insC mutation, the transcriptome data of which were available from public resources.

Patients included in our study originate from Baltic, Caucasus and Volga regions – territories separated by the large distances in European scale. Spanish data is derived from the patients found in nearly the same area.  Thus, we do indeed compare Russian heterogeneous group with a relatively uniform Spanish group. Nevertheless, we managed to uncover common patterns of gene expression that may be inherent to RDEB in general. We believe that the researchers who deal with this rare disease will appreciate these data.

Point 3: They should perform a microarray analysis on their own group”.

Response 3: The suggestion of the Reviewer to perform the microarray analysis is perfectly reasonable. However, we must admit that it is not possible to perform this type of analysis due to the restrictions of the current global pandemic, namely the long time required for the sample delivery from Russia. We believe that RT-qPCR analysis performed on the wider panel of cell lines is an adequate alternative, especially that we conformed the results to be in accordance with the published RDEB transcriptome. Nevertheless, we think that it should be performed in future, and it will be described in the next paper considering the size of the current manuscript and that the cell line number will increase.

Point 4: IHC staining analysis was performed for only 2 patients”.

Response 4: We agree that the lack of ICH data is a serious drawback of this manuscript and added the ICH staining of skin RDEB patient d2, who has a rare type of RDEB-Inversa subtype. After the revision, the Figure 2 now contains four microphotographs from d6, d2, d3 patients and ICH control respectively. The lack of ICH data is caused by the rarity of the disease and the need to get the informed consent from the patients or their legal representatives for the necessity of this analysis. The patient’s state of health must permit the material to be taken.

Round 2

Reviewer 1 Report

I congratulate the authors for this new version, which with the new data provided to the old manustrite, substantially improves the work.

Author Response

We are very grateful to the Reviewer for substantial input to the final version of the manuscript. 

Reviewer 3 Report

Dear authors,

Thank you for your comments. However, please analyze the discussion once again, it is very long. The added point to discussion are OK.

Author Response

We are grateful to the Reviewer for the critical comments. We have shortened and modified the Discussion. As a result, this chapter has 3100 words instead of 4000. We hope that in the present version this manuscript is not overloaded with the unnecessary details.